# Band conductivity oscillations in a gate-tunable graphene superlattice

Robin Huber[1], Max-Niklas Steffen [2], Martin Drienovsky[1], Andreas Sandner[1], Kenji Watanabe [3], Takashi Taniguchi [4], Daniela Pfannkuche[2], Dieter Weiss [1] & Jonathan Eroms [1]✉

Electrons exposed to a two-dimensional (2D) periodic potential and a uniform, perpendicular magnetic field exhibit a fractal, self-similar energy spectrum known as the Hofstadter butterfly. Recently, related high-temperature quantum oscillations (Brown-Zak oscillations) were discovered in graphene moiré systems, whose origin lies in the repetitive occurrence of extended minibands/magnetic Bloch states at rational fractions of magnetic flux per unit cell giving rise to an increase in band conductivity. In this work, we report on the experimental observation of band conductivity oscillations in an electrostatically defined and gate-tunable graphene superlattice, which are governed both by the internal structure of the Hofstadter butterfly (Brown-Zak oscillations) and by a commensurability relation between the cyclotron radius of electrons and the superlattice period (Weiss oscillations). We obtain a complete, unified description of band conductivity oscillations in two-dimensional superlattices, yielding a detailed match between theory and experiment.

[1] Institute of Experimental and Applied Physics, University of Regensburg, D-93040 Regensburg, Germany. [2] I. Institute of Theoretical Physics, University of Hamburg, Notkestraße 9–11, D-22607 Hamburg, Germany. [3] Research Center for Functional Materials, National Institute for Materials Science, 1-1 Namiki, Tsukuba 305-0044, Japan. [4] International Center for Materials Nanoarchitectonics, National Institute for Materials Science, 1-1 Namiki, Tsukuba 305-0044, Japan. ✉email: jonathan.eroms@ur.de

Artificial crystals, realised by moiré superlattices in heterostructures of 2D materials[1–3] or by imposing a nano-patterned superlattice[4–6] on a 2D electron system (2DES) like graphene, provide the opportunity to study transport characteristics of charge carriers in a periodic potential. Under the influence of such a superlattice it becomes possible to modify the band structure and therefore the electronic properties of 2D materials, leading, e.g., to the recent observation of superconductivity in magic-angle graphene[7]. In perpendicular magnetic fields, superlattice systems exhibit a complex magnetic band structure given by the fractal Hofstadter butterfly energy spectrum[8] which was studied in GaAs-based 2DESs[9] and graphene-based systems at cryogenic temperatures[10–12]. At elevated temperatures, leaving the regime of Landau quantisation, the fine structure of the Hofstadter energy spectrum vanishes but its fundamental skeletal structure remains. Temperature-robust magnetoconductivity oscillations were observed, which were labelled Brown–Zak (BZ) oscillations[13,14], and appear periodic in the inverse magnetic flux per unit cell of the lattice. Krishna Kumar et al. identified those oscillations as a band conductivity effect, but mainly interpret them in terms of quasiparticles residing in the minibands of the magnetic band structure introduced by Brown[15] and Zak[16], without resorting to Landau levels. While this interpretation has its merits, as evidenced by ballistic transport of those quasiparticles[17], a full understanding of BZ oscillations is only possible if the band structure of Landau levels (LL) in a 2D periodic potential is taken into account. To this end, we performed magnetotransport experiments in artificially created 2D superlattices[6,18], in which a periodic potential modulation can be controlled by electrostatic means. This approach affords more flexibility in terms of arbitrary lattice constant, geometry and tunable modulation strength compared to moiré superlattices. In particular, using appropriate gate voltages, we enter the regime of weak modulation potential, where the visibility of BZ oscillations is governed by commensurability (Weiss) oscillations. We thus arrive at a unified description of band conductivity oscillations combining both Brown–Zak and Weiss oscillations (WOs). We show below, both experimentally and theoretically that BZ oscillations as well as WOs reflect the dispersion and internal structure of Landau bands at temperatures much larger than the Landau band separation.

The impact of a 2D periodic modulation at high magnetic fields can be understood in three steps. We first consider the Landau level spectrum of an unmodulated 2DES, then activate the modulation potential in one dimension only, leading to Landau bands, and finally turn on the 2D modulation potential, which further splits each Landau band according to the Hofstadter spectrum. In the following, a square 2D superlattice potential $V(\mathbf{r}) = V_0(\cos(Kx) + \cos(Ky))$ with $K = 2\pi/a$, lattice constant $a$, and modulation amplitude $V_0$ is considered. The modulation potential is assumed to be weak ($V_0 \ll \hbar v_F/l_B$), such that Landau level mixing can be neglected ($l_B = \sqrt{\hbar/(eB)}$ is the magnetic length).

A uniform (unmodulated) 2DES, subject to a strong, perpendicular magnetic field develops the Landau level spectrum, giving rise to the quantum Hall effect. For single-layer graphene, due to its linear dispersion, the spectrum has a square root dependence on $B$[19]

$$E_{0,n} = \text{sgn}(n) v_F \sqrt{2\hbar eB|n|}, \qquad (1)$$

where $n$ is the index of the highly degenerate Landau levels, and $v_F$ the Fermi velocity in graphene.

When a 1D modulation potential (for example, the $\cos(Kx)$-term only) is included, each Landau level will broaden into Landau bands, whose width not only depends on the

modulation potential strength, but also on the modulation period and the magnetic field value. The periodic potential in the x-direction leads to a dispersion of the Landau bands with respect to the wave vector in the y-direction, $k_y$, associated with a group velocity $v_{gr} = (1/\hbar)dE_n/dk_y$. The band width, and thus $v_{gr}$, vanish completely at the flat band condition, which, in the semi-classical limit (large $n$)[20], can be described by a commensurability relation between the cyclotron radius of electrons $R_c = \hbar\sqrt{\pi n_S}/(eB)$ and the superlattice period $a$:[21]

$$2R_c = \left(\lambda - \frac{1}{4}\right)a \qquad \lambda = 1, 2, 3, \ldots \qquad (2)$$

This expression contains a dependence on the carrier density $n_S$, and describes the minima of the WOs in the magnetoresistance of a modulated 2DES[21–24]. For single-layer graphene, the full quantum mechanical expression for the Landau band width $\Delta E_n$ was calculated by Matulis and Peeters[25]:

$$\Delta E_n = \frac{V_0}{2} e^{-u/2} \left[L_n(u) + L_{n-1}(u)\right] \qquad (3)$$

Here, $u = K^2 l_B^2/2$, and $L_n(u)$ is a Laguerre polynomial.

Last, we consider the full 2D modulation, for rational values of inverse magnetic flux per unit cell, $\phi_0/\phi = q/p$, with $q$ and $p$ coprime integers. Now, each Landau band, which, for a 1D modulation, depends on $n$ and $k_y$, is split up into $p$ subbands, according to the Hofstadter spectrum in the high-field limit[26]. The overall modified Landau level spectrum is given by:

$$E_n = E_{0,n} + \frac{\Delta\epsilon_\alpha}{2} \cdot \frac{V_0}{2} e^{-u/2} \left[L_n(u) + L_{n-1}(u)\right] \qquad (4)$$

Here $\Delta\epsilon_\alpha$ corresponds to the fractal Hofstadter spectrum at the given value of $\alpha = \phi_0/\phi = q/p$ (see Fig. 1b). The complete situation is sketched in Fig. 1 in which the energy spectrum of Dirac fermions in a 2D square lattice is shown in panels (a) and (c). Due to the linear Dirac dispersion, the Landau levels display a square root dependence on the magnetic field. Each Landau level is first broadened into energy bands, whose width has consecutively

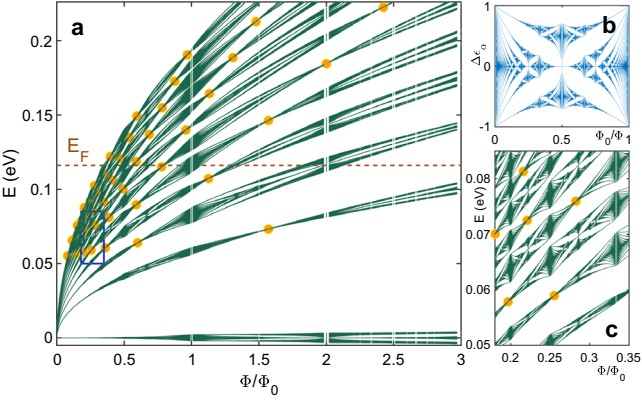

**Fig. 1 Energy spectra of graphene subject to a 2D modulation. a** Modified Landau level spectrum in 2D modulated graphene with $a = 40$ nm and $V_0 = 12$ meV. Landau bands exhibit an internal structure given by the Hofstadter butterfly energy spectrum. Here, only Landau bands with $n \geq 0$ up to $n = 11$ are shown; Landau levels with $n < 0$ can be obtained by reflection at $E = 0$. At unit fractions of the magnetic flux quantum $\phi_0$ per superlattice unit cell the spectrum exhibits extended minibands across the full Landau band width giving rise to an enhanced band conductivity contribution analogous to the case of a 1D modulation. The circles mark the flat band positions at which the usual Landau levels are restored. **b** Hofstadter energy spectrum in the high-field limit, where the inverse flux is the relevant quantity. **c** Zoom into the blue box region of **a**, circles mark the flat band conditions.

more nodes (flat bands) with increasing Landau level index, and then splits up into recursive subbands of the Hofstadter spectrum. Flat band conditions are marked by circles and show the vanishing band width in each Landau band.

The Landau band structure, as shown in Fig. 1a, can be probed in transport experiments. Such experiments, both in GaAs[9] and graphene-based 2DESs[10–12], were performed at low temperatures, where the density of states can be resolved in magnetotransport, and the Hofstadter spectrum can be directly observed, due to the scattering contribution to conductivity. However, the group velocity $v_{gr}$ within each Landau band also leads to a conductivity contribution, the band conductivity $\Delta\sigma_{bc} \propto v_{gr}^2$, which is largest whenever the bands are broad and vanishes at the flat band conditions. In contrast to the density of states effect, band conductivity oscillations survive to higher temperatures, and give rise to the well-known and robust WOs, which were observed both in 2DEGs with a parabolic dispersion[21] and graphene[24,27] for a 1D modulation potential. In a 2D modulated GaAs-based 2DES, the amplitude of WOs was reduced, and this was interpreted as a first indication of the gapped Hofstadter energy spectrum[28]. The fractal gaps split the Landau bands into subbands which reduce the band dispersion and thus also $v_{gr}$, and ultimately the band conductivity. Importantly, the thermal smearing at higher $T$, larger than the Landau level separation, does not restore the band conductivity of the band without gaps[28]. As we will show below, band conductivity oscillations due to the Hofstadter spectrum (Brown–Zak oscillations) can only be observed outside the flat band condition.

The investigated graphene superlattice device with square lattice symmetry and lattice constant $a = 39$ nm was prepared following our previous works[6,18,24] (see "Methods" for more details). By combining one global back gate and a few-layer graphene patterned bottom gate (PBG) a periodic charge carrier density modulation is induced in monolayer graphene encapsulated between two hBN flakes and placed on top of the double gate structure (see Fig. 2b). The back gate mainly tunes the potential modulation strength and the PBG mainly controls the overall charge carrier density in the system. No moiré features were found in this sample, showing that there was no unintentional alignment of graphene and hBN. Data from a second device with a hexagonal superlattice and a coexisting moiré lattice are shown in the Supplemental Material, Fig. S5. In this sample, BZ oscillations in a gate-defined superlattice are confirmed, and they coexist with BZ oscillations due to the moiré lattice.

The realisation of superlattice phenomena by means of our double gating technique is illustrated in Fig. 2c which shows PBG voltage sweeps at three different back gate voltages $V_{bg} = 70, 10,$ and $-70$ V. The inset of Fig. 2c shows the gate map of the device in which the longitudinal resistance $R_{xx}$ is plotted as a function of back gate voltage $V_{bg}$ and PBG voltage $V_{pbg}$ at a temperature of $T = 1.5$ K. The field effect mobility, extracted at low back gate voltage, is about 40,000 cm$^2$ V$^{-1}$ s$^{-1}$. By increasing the modulation strength, mainly controlled by the back gate voltage $V_{bg}$, additional Dirac peaks start to occur due to the induced 2D periodic potential modulation and subsequent band structure modifications. Further, detailed transport data at low temperatures are reported in ref. [6] and the relevant figures are reproduced in the Supplemental Material, for convenience. In particular, at each satellite Dirac peak, the transverse resistance $R_{xy}$, measured at $B = 200$ mT changes sign, which confirms the change of carrier type, when the Fermi level is moved through the minibands (see Supplemental Fig. S1e). We note in passing that the visibility of Hofstadter features in the low-temperature data is highest in regions where the Landau bands are broad (see Supplemental Material, Fig. S6). Upon raising the temperature, the well pronounced satellite Dirac peaks start to vanish in transport

measurements at zero magnetic field, as can be seen in Fig. 2d. In contrast, in magnetotransport measurements (shown in Fig. 3), clear superlattice induced features remain visible even at high temperatures. The following magnetotransport measurements were conducted at a temperature of $T = 125$ K. Due to the lattice constant of $a = 39$ nm, one magnetic flux quantum $\phi_0 = h/e$ threads the superlattice unit cell area already at a magnetic field of about $B = 2.7$ T. As a consequence, in our device it is also possible to study the regime of several magnetic flux quanta at moderate magnetic fields accessible with standard cryostat lab magnets while for moiré superlattices magnetic fields exceeding 50 T would be necessary. The latter are out of scope for static fields even in dedicated high-field facilities. Figure 3a shows the measured longitudinal resistance $R_{xx}$ as a function of magnetic flux per superlattice unit cell area in units of the magnetic flux quantum $\phi/\phi_0$ for several PBG voltages $V_{pbg}$ at a constant back gate voltage of $V_{bg} = 80$ V. In this regime, the Landau quantisation is not resolved due to thermal broadening. Resistance peaks at rational fractions of the magnetic flux quantum are visible, most pronounced at $\phi/\phi_0 = 1$. The positions of the resistance maxima are independent of the applied PBG voltage $V_{pbg}$, i.e., independent of charge carrier density, which is a characteristic of BZ oscillations as they reflect the periodicity of the superlattice only. In addition, one can also observe a distinct feature at two magnetic flux quanta extending over a limited PBG voltage range, but no feature is found for three or four flux quanta, giving further insight into the magnetic band structure. Figure 3b shows the corresponding transverse resistance $R_{xy}$. Similar to the $R_{xx}$ data, the transverse resistance also shows deviations from the straight line behaviour at certain flux ratios.

In order to study the observed features in more detail, following Krishna Kumar et al.[13], we take the second derivative of the conductance which effectively removes the background and highlights extrema. The conductance $G \equiv G_{xx}$ is calculated from $R_{xx}$ and $R_{xy}$ using $G_{xx} = \frac{R_{xx}}{R_{xx}^2 + R_{xy}^2}$ and it is equal to the longitudinal conductivity $\sigma_{xx}$ up to a geometrical factor close to one. In Fig. 4a the second derivative of the conductance $d^2G/dB^2$ is plotted as a function of PBG voltage $V_{pbg}$ and magnetic flux $\phi/\phi_0$ measured at a constant back gate voltage of $V_{bg} = 80$ V. This gate voltage creates a strong modulation potential (see Fig. 2). At rational fractions of the magnetic flux quantum (highlighted in Fig. 4a) clear signatures of BZ oscillations are visible which are most pronounced in the bipolar regime (roughly, where $V_{bg}$ and $V_{pbg}$ have opposite polarity). Also, the feature at two magnetic flux quanta can be observed in a limited $V_{pbg}$ range. At higher charge carrier density, also higher-order states[14] at $\phi/\phi_0 = \frac{2}{3}, \frac{3}{2}$ and weak signatures at $\phi/\phi_0 = \frac{5}{4}, \frac{3}{2}$ appear. By inverting the sign of the applied back gate voltage, the visible features are mirrored at the charge neutrality point (see Fig. 4b for $V_{bg} = -80$ V). By decreasing the modulation strength, i.e., by decreasing the back gate voltage $V_{bg}$, the observed band conductivity oscillations reveal their internal structure. Figure 4c shows data at a back gate voltage of $V_{bg} = 10$ V. The red lines display the flat band condition for $\lambda = 1, 2, ..., 6$. In general, a smaller modulation strength causes a decrease in Landau band width and therefore the effect of extended minibands and the contribution of band conductivity to the overall conductivity is reduced and only the most developed features survive, e.g. at $\phi/\phi_0 = 1$. In addition, compared to stronger potential modulation, an overlap of adjacent Landau bands is reduced and a more sensitive dependence of the conductivity on the oscillatory behaviour of the band width of single Landau bands as a function of the magnetic field can be expected. This effect is most pronounced whenever the flat band condition is fulfilled and the band width approaches its minimum. In experiment this manifests as suppressed band conductivity and

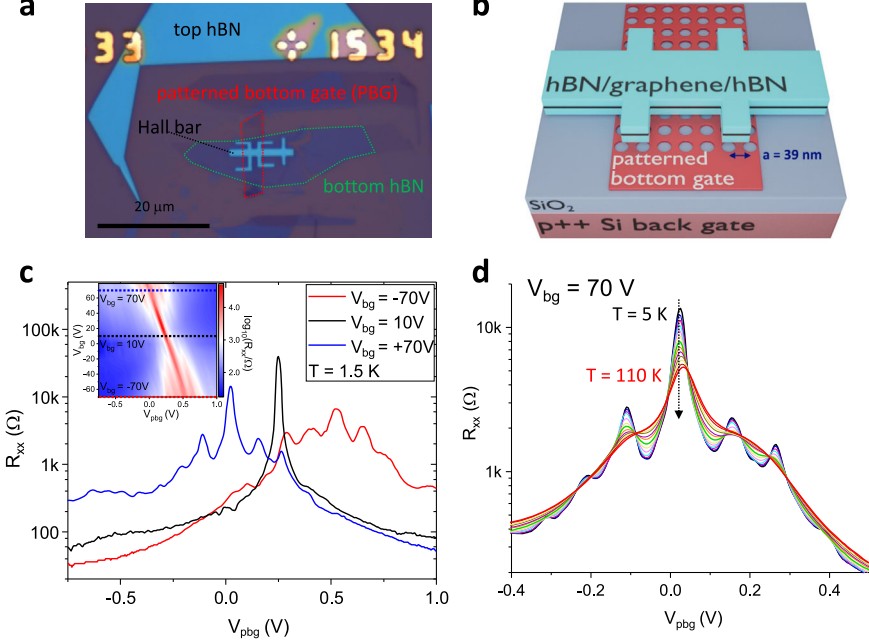

**Fig. 2 Sample layout and basic transport characterisation. a** Optical micrograph of the sample after van der Waals stacking and mesa etching.
**b** Schematic view of the sample design showing the patterned bottom gate underneath the hBN encapsulated graphene sheet. **c** Longitudinal resistance $R_{xx}$ as a function of $V_{pbg}$ at three different back gate voltages $V_{bg}$ at $T = 1.5$ K. By increasing the modulation strength ($V_{bg} = \pm 70$ V), satellite Dirac peaks start to occur. The inset shows the overall gate map of the system, $R_{xx}$ as a function of $V_{pbg}$ and $V_{bg}$. **d** Longitudinal resistance $R_{xx}$ as a function of $V_{pbg}$ at $V_{bg} = 70$ V for different temperatures ranging from $T = 5$ K to $T = 110$ K. By increasing the temperature, superlattice induced satellite Dirac peaks start to vanish.

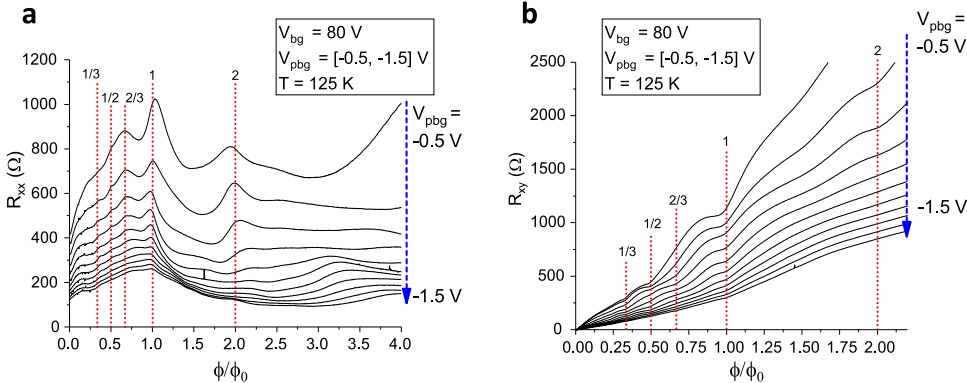

**Fig. 3 Brown–Zak oscillations at different carrier densities.** Longitudinal resistance $R_{xx}$ (**a**) and transverse resistance $R_{xy}$ (**b**) as a function of magnetic flux per superlattice unit cell area $\phi/\phi_0$ for different patterned bottom gate voltages $V_{pbg} = -0.5$ V ... $-1.5$ V at a back gate voltage of $V_{bg} = 80$ V. At rational fractions of the magnetic flux quantum, clear resistance peaks in $R_{xx}$ are visible accompanied by dips in the transverse resistance with the most pronounced feature at $\phi/\phi_0 = 1$.

can be observed in the data in Fig. 4c at $\phi/\phi_0 = 1$. This also becomes apparent in line cuts of the raw $R_{xx}$ data (see Fig. 4d). Around $\phi/\phi_0 = 1$, a BZ peak is visible, but only in between flat band conditions, which are marked by red lines. Conversely, WOs are visible in each trace, with their minima at the flat band positions, but they only appear clearly at the positions where the BZ oscillations show a maximum, thus modulating the BZ feature at $\phi/\phi_0 = 1$. Following from this also the feature at $\phi/\phi_0 = 2$ at high back gate voltage (see, e.g., Fig. 4b) which is localised in a certain PBG voltage range can be explained as it appears exactly between two flat band positions with $\lambda = 1$ and $\lambda = 2$ (see also Supplemental Fig. S4b). This highlights the importance of both the oscillating width of the Landau bands and their internal structure due to the Hofstadter spectrum in the interpretation of the underlying physics. We note that WOs are usually described

in the regime of weak modulation, but they still remain visible at stronger modulation potential, see, e.g., the non-vertical ridges in Fig. 4b running parallel to the flat band conditions. At this strong modulation, Landau bands overlap and the visibility of BZ oscillations is only weakly affected by the flat band condition. Note that strong modulation is present in particular in the bipolar region. In the unipolar region, we can also find the weak modulation regime for $V_{bg} = \pm 80$ V and a range of $V_{pbg}$. This is seen more clearly on an adapted gray scale (see Supplemental Fig. S4). Strong modulation is also present in moiré superlattices, where the potential cannot be tuned to the weak regime, and BZ oscillations therefore show no signs of WOs. The visibility of BZ oscillations is not affected by the overlap of Landau bands, as bands and gaps in the Hofstadter spectrum depend on the flux per unit cell, but not the Landau level index.

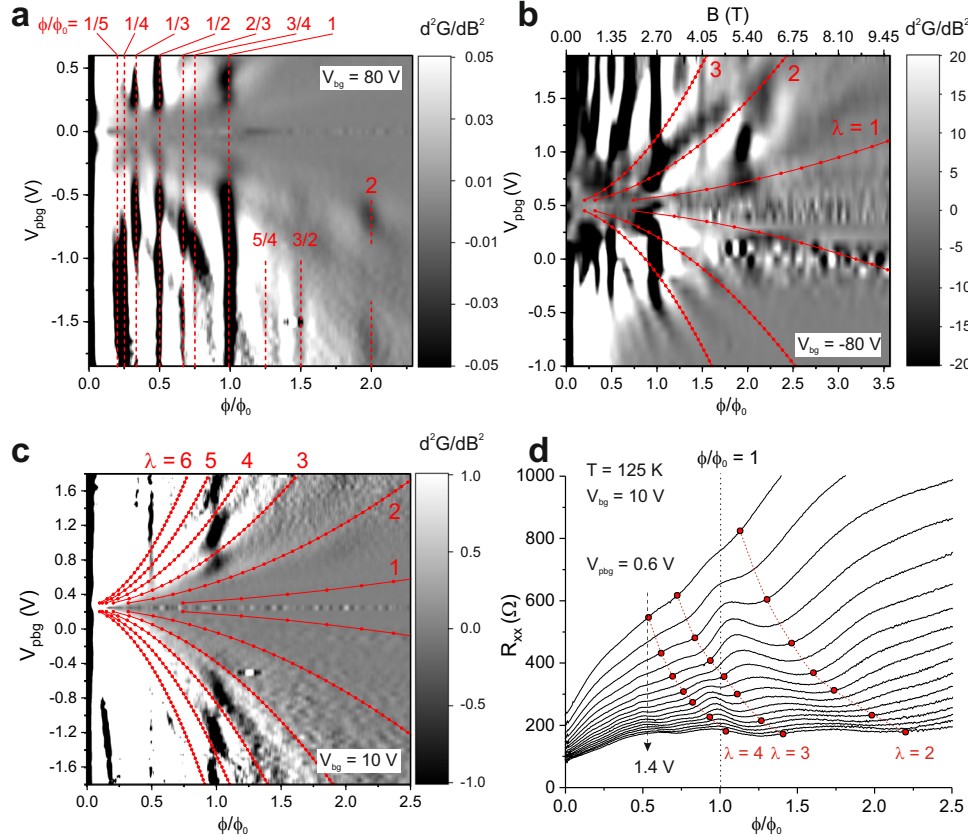

**Fig. 4 Coexistence of Brown–Zak (BZ) and Weiss oscillations (WO). a** Gray scale plot of the second derivative of the conductance $d^2G/dB^2$ as a function of magnetic flux $\phi/\phi_0$ and patterned bottom gate voltage $V_{pbg}$ at a back gate voltage $V_{bg} = 80$ V. Band conductivity/Brown–Zak oscillations are mainly visible in the bipolar regime. Observed features at rational fractions of the magnetic flux quantum are highlighted with red dashed lines and labelled with the corresponding value of $\phi/\phi_0$. Also, weak features at $\phi/\phi_0 > 1$ can be observed. **b** Gray scale plot of $d^2G/dB^2$ as a function of $\phi/\phi_0$ and $V_{pbg}$ at $V_{bg} = -80$ V. By inverting the polarity of the back gate voltage the features are mirrored with respect to the charge neutrality point. The localized feature at $\phi/\phi_0 = 2$ lies between flat band positions with $\lambda = 1$ and $\lambda = 2$. **c** Gray scale plot of $d^2G/dB^2$ as a function of $\phi/\phi_0$ and $V_{pbg}$ at $V_{bg} = 10$ V. Reduction of band conductivity oscillations at smaller back gate voltage, i.e. weaker potential modulation. A suppression of the most pronounced feature at $\phi/\phi_0 = 1$ can be observed whenever the flat band condition for $\lambda = 1, 2, ..., 6$ is fulfilled. **d** Longitudinal resistance $R_{xx}$ at $V_{bg} = 10$ V and $V_{pbg} = 0.6$ V...1.4 V in 0.05 V steps. In contrast to Fig. 3 (a), the BZ features are much weaker. Instead, WO are visible, governed by the flat band conditions (given by red dots; red lines are guides to the eye), but only appear clearly at BZ maxima positions, modulating the BZ feature at $\phi/\phi_0 = 1$.

The experimental observations can be modelled and accurately reproduced by considering band conductivity corrections due to the periodic potential, leading to oscillations caused by both the commensurability between cyclotron diameter and lattice period (Weiss oscillations, following Eq. (2), see Fig. 5a) and by the varying number and width of subbands within the Hofstadter spectrum (see Fig. 5b). The band conductivity correction due to a superlattice potential is calculated from the Kubo formula, and is therefore proportional to the square of the band velocity, which in turn is proportional to the band width.

Once level broadening by impurity scattering is small enough to allow (partially) resolving the Hofstadter spectrum, the gaps in the Hofstadter spectrum reduce the Landau band width below its value obtained from a 1D modulation (Eq. 3). This also reduces the band velocity, and, as the conductivity is proportional to the velocity squared, it is still reduced even after summation over the $p$ Hofstadter subbands. Qualitatively, this suppresses the strong band conductivity oscillations when going from a 1D modulation to 2D modulation[28,29] (see Fig. 5a, b). On the other hand, for stronger collision broadening, subband splitting does not lead to a reduction of the band conductivity oscillations (except for shorter electron mean free path).

The experimental observations in our devices can now be understood by considering Fig. 1 once more. For integer values of inverse flux, $\phi_0/\phi$, in other words unit fractions of the flux $\phi/\phi_0$, the Hofstadter spectrum has the full width of the underlying Landau bands[8,26] (See Fig. 1b at $\phi_0/\phi = 0$ or 1). Therefore, the band conductivity is as large as permitted by the modulation broadened Landau bands. At other inverse flux values $\phi_0/\phi = q/p$ with small $p$, e.g. 1/3 or 2/3, there are still sizable subbands in the Hofstadter-split Landau bands, which are also reflected as visible conductivity contributions. Outside those regions, the Hofstadter spectrum is so strongly split that band conductivity is completely suppressed. As the Hofstadter spectrum only depends on the flux ratio, not on the density, all vertical dark lines in Fig. 4 can be traced back to this effect. To understand the density dependent modulation in the vertical lines (in particular, the $\phi/\phi_0 = 1$ line in Fig. 4c), we have to consider the flat band condition. Depending on the flat band condition, the width of each Landau band oscillates with magnetic field, leading to an oscillating band conductivity correction[23]. For example, in Fig. 1a, we find two Landau bands ($n = 6, 11$), where the flat band condition occurs very close to $\phi/\phi_0 = 1$, reducing the visibility of the corresponding conductivity peak, while other Landau bands extend to the maximum width. We note that the semi-classical formula for the flat band condition (Eq. 2) is sufficient to describe the data in Fig. 4. Clearly, the visibility of the BZ oscillations is suppressed, whenever the flat band condition is satisfied.

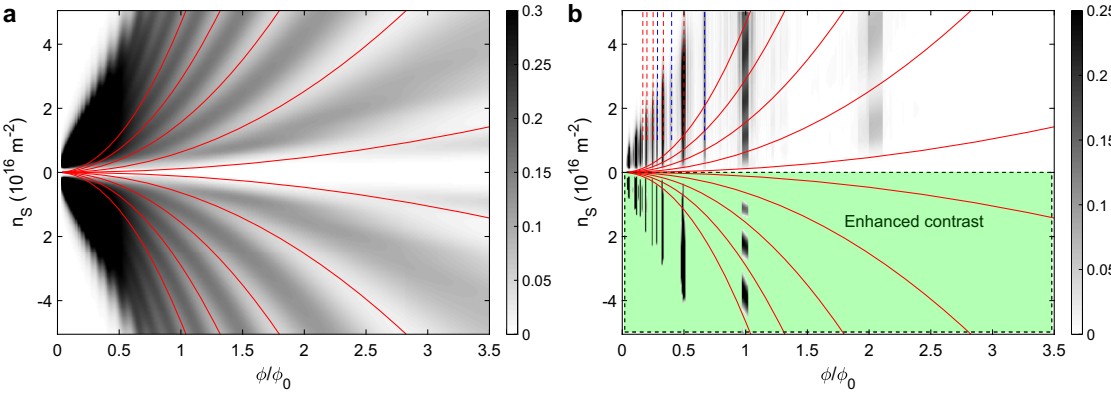

**Fig. 5 Gray scale plots of the calculated conductivity contribution, in arbitrary units. a** For a 1D superlattice, band conductivity oscillations appear, which depend on both the carrier density $n_S$ and flux quanta per unit cell $\phi/\phi_0$. **b** In the case of a 2D superlattice, the band conductivity oscillations of the 1D case are mostly suppressed (light regions), except whenever the Hofstadter butterfly has extended regions (dark bands), at rational fractions of $\phi/\phi_0$. The features in the experimental data are reproduced, including the appearance of band conductivity maxima at rational fractions of $\phi/\phi_0$ and the modulation of the observed Brown–Zak oscillations by Weiss oscillations (lower part: enhanced contrast). The red continuous lines in both figures show the semi-classical flat band condition (see Eq. 2). Dashed lines mark rational $\phi/\phi_0 < 1$, red for unit fractions, blue other. Note that the two triangular regions at low $\phi/\phi_0$ and high $|n_S|$ do not contain meaningful data due to a limited number of Landau levels in the calculation.

Using the approach outlined above, with more calculation details given in the "Methods" section, we obtain a semi-quantitative estimate of the band conductivity. The result is plotted in Fig. 5b, which reproduces the main experimental features. Most importantly, the band conductivity picture reproduces the Brown–Zak oscillations, which appear as dark vertical lines at rational fluxes. Clearly, by considering the subband structure of the Hofstadter butterfly, BZ oscillations are well described. We also confirm that unit fractions of the flux per unit cell ($\phi/\phi_0 = 1/q$) lead to the strongest features and weaker vertical lines appear for other fractions, as explained above. In particular, as in experiment, the feature at $\phi/\phi_0 = 2$ is much weaker, and no feature is found for $\phi/\phi_0 = 3$. Owing to commensurability oscillations, the vertical lines at $\phi/\phi_0 = 1$ and $\phi/\phi_0 = 2$ are visibly modulated, following Eq. (2), but this modulation is absent for $\phi/\phi_0 \ll 1$ due to thermal broadening. The latter fact is more clearly visible in a 3D plot of the same data, which we show in the Supplementary Information, Fig. S7.

In conclusion, we present band conductivity oscillations in an artificial, electrostatically defined, and gate-tunable 2D graphene superlattice. Band conductivity oscillations of the Brown–Zak type are clearly visible for one and two flux quanta per unit cell and several orders of fractions of flux quanta. In addition, at sufficiently weak superlattice potential, those oscillations reveal their internal structure determined by Weiss oscillations, and partly vanish whenever a commensurability condition between the cyclotron radius of the electrons and the superlattice period is fulfilled. Our transport measurements and theoretical description provide new insight into the magnetic band structure of graphene superlattices and we show that the manifestation and experimental visibility of superlattice induced features and band conductivity oscillations depends crucially on the occurrence and width of energy bands in the magnetic band structure.

## Methods

**Sample fabrication and data acquisition**. Few-layer graphene was exfoliated from natural graphite onto oxidised silicon wafers and patterned by standard electron beam lithography (EBL) and reactive ion etching (RIE) with $O_2$ plasma to form the patterned bottom gate (PBG). After etching, the PBG was cleaned in Remover PG (Microchem) at 60°C and annealed in vacuum at 400 °C in order to remove resist residues. A standard van der Waals pick up technique[30] was used to encapsulate monolayer graphene between two hBN flakes and to transfer the hBN/graphene/hBN stack onto the PBG. hBN was exfoliated from bulk, high-quality hBN crystals[31], and single-layer graphene from natural graphite. Flakes were picked up

and assembled using a polydimethylsiloxane/polycarbonate stack on a microscope glass slide, in a custom made holder in an optical microscope, whose x-y-stage served to position the flakes to a precision of about 1 μm. The sample described in the main text shows no sign of a moiré pattern, while in a second sample, the crystal axes of graphene and one hBN crystal were unintentionally aligned, leading to well-defined moiré superlattice features. The bottom hBN layer had a thickness of only ~5 nm in order to obtain a well-defined potential modulation. The final stack was etched into Hall bar geometry by EBL and RIE with $SF_6$[32] and $O_2$, and edge contacts[30] were fabricated by EBL and evaporation of Cr (5 nm)/Au (80 nm). All transport measurements were conducted in a $^4$He cryostat with standard lock-in techniques at a source current of 10 nA. Data acquisition was done via the Lab::Measurement environment[33].

**Calculations**. For graphene with a one-dimensional superlattice modulation, the following band conductivity correction is obtained[25]:

$$\sigma_{bc,1D} \propto u\,e^{-u} \sum_{n=0}^{\infty} \frac{g(E_{0,n})}{[g(E_{0,n})+1]^2}\left[L_n(u) + L_{n-1}(u)\right]^2 \qquad (5)$$

with $g(E) = \exp((E - E_F)/(k_B T))$, and $E_{0,n}$ the unperturbed Landau level energy. This gives rise to the Weiss oscillations in a 1D superlattice, and also modulates the visibility of BZ oscillations in the 2D superlattice.

To obtain a semi-quantitative estimate of the band conductivity in a 2D superlattice[28,29], we first calculate the correction due to a 1D potential, following Eq. (5), where we include all Landau levels within $\pm 10 k_B T$ of the Fermi level. For numerical stability, we restrict the maximum number of Landau levels to 30, which affects the regions with $\phi/\phi_0 < 0.5$ and high carrier density. For each magnetic field, we then obtain the Hofstadter spectrum, and broaden it to only retain gaps exceeding a minimum size, thus mimicking collision broadening. The reduced band width due to Hofstadter splitting is taken into account to reduce the overall conductivity. As the group velocity in each miniband depends on its width, and the band conductivity is proportional to the square of the group velocity, we sum over the square of the width of all Hofstadter subbands within each Landau band to scale down the conductivity value obtained from Eq. (5).

## Data availability

The Experimental Data used in this study are available upon request in the electronic publication database of the University of Regensburg under https://doi.org/10.5283/epub.51676.

## Code availability

The Matlab Code used in this study is available upon request in the electronic publication database of the University of Regensburg under https://doi.org/10.5283/epub.51676.

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

## Acknowledgements

The authors gratefully acknowledge funding by the Deutsche Forschungsgemeinschaft (DFG, German Research Foundation) SFB 689, project B5 (A.S.) project-id 314695032 (SFB 1277, subproject A09) (R.H.); project-id 426094608 (ER 612/2-1) (J.E.), WE 2476/11-2 (R.H., D.W.) and GRK 1570 (R.H., M.D., A.S.). This project has received funding from the European Research Council (ERC) under the European Union's Horizon 2020 research and innovation programme (grant agreement No 787515) (R.H.). K.W. and T.T. acknowledge support from the Elemental Strategy Initiative conducted by the MEXT, Japan, Grant Number JPMXP0112101001, JSPS KAKENHI Grant Number JP20H00354 and the CREST(JPMJCR15F3), JST.

## Author contributions

R.H., D.W., and J.E. designed the experiment, R.H. performed sample fabrication and transport measurements and processed the experimental data, M.D. and A.S. contributed to the fabrication procedures, K.W. and T.T. grew the hBN crystals, M.-N.S. and D.P. provided data of the Hofstadter spectrum, J.E. performed transport calculations. R.H., J.E. and D.W wrote the manuscript with contributions from all authors.

## Funding

## Competing interests

The authors declare no competing interests.
