## [Peer Review File · Nature Communications]

Reviewers' Comments:

Reviewer #1:

Remarks to the Author:

In this work Robin Huber et. al. study the properties of artificial graphene-based superlattices with the main focus on electron transport in a high magnetic field. They compare the fan diagrams measured for different superlattice modulation strengths and claim the observation of the 'Weiss' oscillations in addition to the Brown-Zak (BZ) oscillations which become visible for the small superlattice modulation strength.

I do not recommend this manuscript for publication in Nature Communications. My main problem is related to the experimental part of this work. It is not convincing.

1) The main result of this work is presented in Fig. 4 where the authors show the second derivative of longitudinal conductivity by the magnetic field. For modulation strength $V_{bg}=10V$ (Fig. 4d) they highlight the expected positions of the Weiss oscillations which agree with some perturbation of BZ oscillations observed in their measurements. But most of the lines don't correspond to anything. The authors also send the reader to see Fig. S3, but in fact, it doesn't help at all. One can see that when $\lambda = 4$ crosses flux 1 the resistance shows small maxima, while when $\lambda = 3$ crosses flux 1 the resistance shows minima.

2) As I understand authors claim that Weiss oscillations should become pronounced for the weak superlattice modulation potential. But in Fig. 4a and Fig. 4c where they show data for the strong modulation potential, one can also see the multiple perturbations of BZ oscillations which look similar to their measurements shown in Fig. 4d.

3) The main conclusions of this work are made with a single device. There is one more device shown in Fig. S1 but it doesn't show any signatures of Weiss oscillations and doesn't support the main claim at all.

Also it feels like the authors simply try to recycle their old data and didn't try to measure specifically for this paper. For example, for the proper information about the sample (sample dimensions, fabrication etc.) they simply send the reader to their old works and don't show the information in this work. Any low-temperature fan diagram measurements from their device should be also searched and guessed in their old works. All these could be presented for example in the supplementary materials.

Reviewer #2:

Remarks to the Author:

Review on "Brown-Zak and Weiss oscillations in a gate-tunable graphene superlattice: A unified picture of miniband conductivity"

The authors report an interesting behaviour of artificial graphene superlattices revealed by magnetotransport experiments and supported by extensive theoretical consideration. The superlattices are electrostatically-defined and gate-tunable that allowed the authors not only to probe the effect of external periodic modulation on the electronic properties of graphene but also to control the modulation strength which until recently was truly challenging in graphene superlattices (e.g. of moiré type). Such a powerful knob enabled the authors not only to observe the presence of Brown-Zak oscillations in such superlattices but also to reveal their intricate interplay with Weiss oscillations occurring close to the flat band regions of the energy spectrum. For example, modification of the oscillations close to $\nu/\nu_0=1$ and their strong suppression close to the $\nu/\nu_0=2$ fillings, supported by theoretical analysis, is a rather non-trivial effect not addressed previously. Overall, the results of this work are interesting and I believe are important for the 2D community dealing with superlattices. Therefore I would recommend the publication of these results if the authors address the following minor issues:

1. All the data presented in the main text is related to longitudinal measurements. However, I

believe it can also be interesting to explore R_{xy} and σ_{xy} behaviour. Can the authors present such data? The 3D render of the device shows a Hall bar, that is why I presume multi-terminal measurements were possible, including the R_{xy} configuration. I believe some of the features could be more visible on σ_{xy} maps than on those obtained by the differentiation. In particular, the authors refer to the satellite peaks in Fig.1 as superlattice neutrality points where R_{xy} should change sign. Can the authors demonstrate this?

2. The authors may want to specify how the conductance, G , is obtained. Is it simply $1/R_{xx}$? If so, why do the authors use $\Delta \sigma_{bc}$ in the text as well? Was R_{xx} obtained in the four-terminal configuration? I believe this inconsistency has to be corrected.

3. Fig. 3b, in my opinion, is absolutely not informative. Red-white-blue gradients in this case do not carry any information and that is why I would encourage the authors to remove this map from the text as it may even confuse some readers. Fig. 3a is self-sufficient.

4. On the contrary, I think Fig. S3 has to appear in the main text because it is the raw data demonstrating where the flat band points occur and what effect the latter have on band oscillations.

5. What are the sample dimensions? A photograph of the sample could bring some clarity in the sample design.

6. Could the authors also comment on the possibility to observe similar effects in more conventional moiré graphene superlattices.

Overall, I believe the paper is well written and will find its broad readership among Nat. Comm. community.

Reviewer #3:

Remarks to the Author:

The authors provide a comprehensive description based on their theoretical and experimental results of the band structure of graphene concomitantly subjected to a periodic potential and perpendicular magnetic field. The observation of Hofstadter butterfly in moiré superlattices of the graphene /h-BN system has been reported almost a decade ago. More recently, the observation of conductance oscillations persisting at high temperatures renewed interest in the field. In this work, the superlattice potential is realized through electrostatic gating, enabling the authors to tune the strength of the potential, which is absent in moiré superlattices. Exploiting this additional degree of freedom, tuning the system into the weak potential modulation regime, enables them to clearly resolve the internal structure (Weiss oscillations) of the Brown-Zak conductivity oscillations.

In my judgement, this work is a highly comprehensive description of the band structure and conductance oscillations observed in graphene superlattices subjected to external magnetic fields. The results presented here are not particularly surprising, but the study could be very useful for providing a general picture for the interpretation of a series of experimental findings found in the literature, and also corroborated by the authors own experimental results.

I also have the following questions:

- Using h-BN encapsulated graphene raises the question of possible (in this case unwanted) additional superlattice potential that could complicate the interpretation of the results. Did the authors ensure that the rotation of graphene relative to h-BN is high enough to avoid the perturbation of the electron system in the range probed by transport measurements?

- What is the reason for the anisotropy regarding the carrier density of the $\nu = 2$ state? In the experimental case, there can be many sources of asymmetry, but since it appears clearly also in calculations (fig. 5b) it should be identifiable.

For ease of comparison, we enclose a pdf with all deleted parts of the text crossed out and new parts in color blue (prepared by latexdiff). Page numbers in the replies correspond to this version. Figs 2 through 5 were replaced with revised versions.

Point by point reply to the reviewers' comment:

Reviewer #1 (Remarks to the Author):

In this work Robin Huber et. al. study the properties of artificial graphene-based superlattices with the main focus on electron transport in a high magnetic field. They compare the fan diagrams measured for different superlattice modulation strengths and claim the observation of the Weiss oscillations in addition to the Brown-Zak (BZ) oscillations which become visible for the small superlattice modulation strength.

I do not recommend this manuscript for publication in Nature Communications. My main problem is related to the experimental part of this work. It is not convincing.

1) The main result of this work is presented in Fig. 4 where the authors show the second derivative of longitudinal conductivity by the magnetic field. For modulation strength $V_{bg}=10V$ (Fig. 4d) they highlight the expected positions of the Weiss oscillations which agree with some perturbation of BZ oscillations observed in their measurements. But most of the lines don't correspond to anything. The authors also send the reader to see Fig. S3, but in fact, it doesn't help at all. One can see that when $\lambda = 4$ crosses flux 1 the resistance shows small maxima, while when $\lambda = 3$ crosses flux 1 the resistance shows minima.

In contrast to the referee's impression and statement regarding figure 4d (4c in the revised version), the data show very clearly that the flat band condition for the Weiss oscillations modulates the Brown-Zak oscillations. Maxima of Brown-Zak oscillations appear most clearly for $\Phi/\Phi_0 = 1$ in figure 4c. For this flux a Landau band is not split due to the Hofstadter spectrum and thus the band-conductivity contribution of a broadened Landau band (giving the Brown-Zak maxima) is maximum. But if the Landau bandwidth vanishes – this is the case at the red dashed lines in Fig. 4c, which describe minima positions of the Weiss-oscillation – the Brown-Zak oscillation maximum gets suppressed. This is what is observed at $\Phi/\Phi_0 = 1$ in figure 4c: At the position of the flat band condition (red dashed lines), the black vertical line due to Brown-Zak maxima is interrupted.

The reviewer also criticizes that “most of the lines don’t correspond to anything”. On the contrary, looking at figure 4d (4c in the revised version), one notes that the lines for $\lambda = 2,3,4,5$ mark the positions of the modulations in the Brown-Zak feature at $\Phi/\Phi_0=1$, precisely underscoring our point of a common origin of Brown-Zak and Weiss oscillations. Features for smaller and higher flux are less pronounced. In the case of smaller flux the band width in the weakly modulated case is too small to generate sizeable Brown-Zak oscillations when for fluxes $1/2 \Phi_0$, $1/3 \Phi_0$, $1/4 \Phi_0$ etc. the Landau bands are not split by the Hofstadter spectrum. For larger fluxes the Landau bands are always split by the Hofstadter spectrum causing lower amplitudes of the resistance oscillations, which in this regime reflect the band conductivity within the Hofstadter butterfly.

Regarding Fig. S3 (now Fig 4d of the main text, as suggested by reviewer 2) the observations of the reviewer are correct but need to be seen from another perspective. To best see this, it is useful to go along the $\Phi/\Phi_0 = 1$ line in Fig. 4d (previous S3). This corresponds going along the grey scale plot at $\Phi/\Phi_0 = 1$ between $V_{\text{pbg}} = 0.6 \text{ V}$ and 1.4 V in Fig. 4c. The minima along this line occur in the vicinity of the flat band condition, i.e., the red dots. The maxima correspond here to the Brown-Zak or Weiss oscillation maxima. In fact, if the maximum condition for the Weiss oscillations coincides with $\Phi/\Phi_0 = 1$, Weiss-oscillations and Brown-Zak oscillation amplitude are the same as they result from the same Landau band dispersion and band conductivity. Outside the regions of broad bands in the Hofstadter spectrum (in this case, outside $\Phi/\Phi_0 = 1$) the WOs are strongly suppressed, in accordance to theoretical considerations by Gerhardts et al. (Refs. 28 and 29).

To trace the visibility of the BZ oscillations, one has to take the second derivative of G with respect to B , as introduced by the Manchester group in their papers on BZ oscillations in moiré superlattices, and follow the evolution on a grayscale map, precisely as we did in Fig 4. The fact that the individual resistance trace shows a minimum, where $\lambda = 3$ coincides with $\Phi/\Phi_0 = 1$ is no contradiction: The minimum is expected as a manifestation of Weiss oscillations. The corresponding BZ feature is suppressed, but this only becomes apparent in the gray scale plot.

To remove possible confusion, we moved this figure to the main text and added a corresponding description in the text (caption of Fig 4d and p.3, right column).

2) As I understand authors claim that Weiss oscillations should become pronounced for the weak superlattice modulation potential. But in Fig. 4a and Fig. 4c where they show data for the strong modulation potential, one can also see the multiple perturbations of BZ oscillations which look similar to their measurements shown in Fig. 4d.

The flat band condition for the Weiss oscillations holds precisely only in the limit of weak periodic potentials, i.e., the regime of a weak perturbation to the Landau levels. Therefore, one usually chooses the regime of a weak modulation potential to study them. In the opposite limit of very strong modulation, such as an antidot lattice, commensurability features are also observed, but they follow a slightly different commensurability condition. In the transition between both limiting cases, one can still observe oscillations in magnetotransport, which are governed by a relation containing the cyclotron radius, which has a square root dependence on carrier density. Therefore, it is not surprising that we still see dark and bright lines appearing in the gray scale plot, which run parallel to the flat band condition, and appear throughout the plot. Given that the periodic potential is too strong, we cannot give the expected minima positions as a closed form expression, as for the two limiting cases, but commensurability oscillations can be present for any modulation strength between weak and very strong. We added a description to this effect to the main text on p.4, left column, first paragraph.

The referee is right in pointing out that the data at large back gate voltage also results in modulation of the BZ feature. This is more apparent at different contrast settings, which we now include in the Supplemental Material, Fig S4 and shows up in the unipolar regime, where the modulation potential is weaker than in the bipolar regime. We added a discussion of this matter on p.4, left column and in Supplemental section II.

3) The main conclusions of this work are made with a single device. There is one more device shown in Fig. S1 but it doesn't show any signatures of Weiss oscillations and doesn't support the main claim at all.

In the sample with moiré and gate-induced superlattice potential we confirm that BZ oscillations are present in a gate-defined superlattice. In contrast to the moiré case, this has not yet been shown in the literature (to our knowledge). Due to the complicated oscillation pattern resulting from two different periods, we decided not to study this sample in the low modulation regime.

Also it feels like the authors simply try to recycle their old data and didn't try to measure specifically for this paper. For example, for the proper information about the sample (sample dimensions, fabrication etc.) they simply send the reader to their old works and don't show the information in this work. Any low-temperature fan diagram measurements from their device should be also searched and guessed in their old works. All these could be presented for example in the supplementary materials.

We regret that the Referee got this impression, the more so as the data in this manuscript were specifically taken for the purpose of studying Brown-Zak oscillations, which are only visible at elevated temperature. Data in our previous publication were taken at 1.5 K only. At low temperature, quantum oscillations dominate the transport data, rendering Brown-Zak oscillations invisible. For the present manuscript, we took several complete data sets at $T = 125$ K, and at several gate voltages. Those two-dimensional color plots are very time consuming. The 125 Kelvin measurements were not published previously.

We also note that it is not uncommon to publish several papers on data taken on the same device. As a prominent example, we refer to the work by Y. Saito, et al, from Andrea Young's group, where data from a single device resulted in at least three journal publications (Nature **592**, 220 (2021) (see remark in Methods section, Device fabrication), Nature Physics **16**, 926 (2020), Nature Physics **17**, 478 (2021)).

We agree with the reviewer that it would be helpful to have characterization data from the previous publication readily available. Therefore, we decided to reproduce the first three figures from our Nano Letters publication in the Supplemental Material, in accordance with ACS's copyright agreement.

Reviewer #2 (Remarks to the Author):

Review on "Brown-Zak and Weiss oscillations in a gate-tunable graphene superlattice: A unified picture of miniband conductivity"

The authors report an interesting behaviour of artificial graphene superlattices revealed by magnetotransport experiments and supported by extensive theoretical consideration. The superlattices are electrostatically-defined and gate-tunable that allowed the authors not only to probe the effect of external periodic modulation on the electronic properties of graphene but also to control the modulation strength which until recently was truly challenging in graphene superlattices (e.g. of moiré type). Such a powerful knob enabled the authors not only to observe the presence of Brown-Zak oscillations in such superlattices but also to reveal their intricate interplay with Weiss oscillations occurring close to the flat band regions of the energy spectrum. For example, modification of the oscillations close to $\nu/\nu_0=1$ and their strong suppression close to the $\nu/\nu_0=2$ fillings, supported by theoretical analysis, is a rather non-trivial effect not addressed previously. Overall, the results of this work are interesting and I believe are important for the 2D community dealing with superlattices. Therefore I would recommend the publication of these results if the authors address the following minor issues:

1. All the data presented in the main text is related to longitudinal measurements. However, I believe it can also be interesting to explore R_{xy} and σ_{xy} behaviour. Can the authors present such data? The 3D render of the device shows a Hall bar, that is why I presume multi-terminal measurements were possible, including the R_{xy} configuration. I believe some of the features could be more visible on σ_{xy} maps than on those obtained by the differentiation. In particular, the authors refer to the satellite peaks in Fig.1 as superlattice neutrality points where R_{xy} should change sign. Can the authors demonstrate this?

We welcome this suggestion and include R_{xy} data in the revised Fig 3b. Also, the low-temperature R_{xy} data from our Nano Letters are shown in Fig S1, where the sign change at the superlattice neutrality points is proven.

2. The authors may want to specify how the conductance, G , is obtained. Is it simply $1/R_{xx}$? If so, why do the authors use $\Delta \sigma_{bc}$ in the text as well? Was R_{xx} obtained in the four-terminal configuration? I believe this inconsistency has to be corrected.

We thank the reviewer for pointing this out. We include the calculation of $G=G_{xx}$ from the four-terminal R_{xx} and the R_{xy} data into the main text (p.3 right column). σ_{xx} is essential equal to G_{xx} , up to a geometrical factor close to one.

3. Fig. 3b, in my opinion, is absolutely not informative. Red-white-blue gradients in this case do not carry any information and that is why I would encourage the authors to remove this map from the text as it may even confuse some readers. Fig. 3a is self-sufficient.

We follow the suggestion of the referee and replaced the color plot by the more informative plot of R_{xy} in the Brown-Zak regime.

4. On the contrary, I think Fig. S3 has to appear in the main text because it is the raw data demonstrating where the flat band points occur and what effect the latter have on band oscillations.

This is helpful advice, we therefore reorganized Fig. 4 to include Fig S3 into the main text.

5. What are the sample dimensions? A photograph of the sample could bring some clarity in the sample design.

The referee is correct, we should have shown a micrograph with scale bar. It is now included in Fig 1a (after mesa etch) and in Fig S1 (with contacts)

6. Could the authors also comment on the possibility to observe similar effects in more conventional moiré graphene superlattices.

To address this point, we expanded the discussion of strong modulation and moiré lattices on p. 4, upper left

Overall, I believe the paper is well written and will find its broad readership among Nat. Comm. community.

We thank the referee for this positive assessment.

Reviewer #3 (Remarks to the Author):

The authors provide a comprehensive description based on their theoretical and experimental results of the band structure of graphene concomitantly subjected to a periodic potential and perpendicular magnetic field. The observation of Hofstadter butterfly in moiré superlattices of the graphene /h-BN system has been reported almost a decade ago. More recently, the observation of conductance oscillations persisting at high temperatures renewed interest in the field. In this work, the superlattice potential is realized through electrostatic gating, enabling the authors to tune the strength of the potential, which is absent in moiré superlattices. Exploiting this additional degree of freedom, tuning the system into the weak potential modulation regime, enables them to clearly resolve the internal structure (Weiss oscillations) of the Brown-Zak conductivity oscillations.

In my judgement, this work is a highly comprehensive description of the band structure and conductance oscillations observed in graphene superlattices subjected to external magnetic fields. The results presented here are not particularly surprising, but the study could be very useful for providing a general picture for the interpretation of a series of experimental findings found in the literature, and also corroborated by the authors own experimental results.

I also have the following questions:

- Using h-BN encapsulated graphene raises the question of possible (in this case unwanted) additional superlattice potential that could complicate the interpretation of the results. Did the authors ensure that the rotation of graphene relative to h-BN is high enough to avoid the perturbation of the electron system in the range probed by transport measurements?

In this work, two samples were studied: The sample presented in the main text had a large rotation angle between hBN and graphene to avoid a moiré lattice, and we do not observe any related features. The additional sample shown in the Supplemental does actually have this undesired moiré superlattice. Therefore we observe BZ oscillations related to two different lattice periods. We now address this point explicitly in the Methods (Sample fabrication...) section of the manuscript and on p.2, right column.

- What is the reason for the anisotropy regarding the carrier density of the $\phi/\phi_0 = 2$ state? In the experimental case, there can be many sources of asymmetry, but since it appears clearly also in calculations (fig. 5b) it should be identifiable.

It seems that the presentation of the data in Fig 5b was misleading. In Fig 5b, in the lower half the contrast was chosen differently to highlight in particular the modulation of the $\phi/\phi_0 = 1$ feature. At this contrast setting, the $\phi/\phi_0 = 2$ feature is completely washed out. There is no anisotropy, it is just a matter of presenting the data. To avoid this confusion, we regenerated Fig 5b, with a light green filled rectangle in the part with enhanced contrast.

Reviewers' Comments:

Reviewer #1:

Remarks to the Author:

During the revision, authors have significantly improved the quality of the manuscript. I need to agree with the authors that my first impression of this work was wrong. Now the paper can be published in Nature Communication.

Reviewer #2:

Remarks to the Author:

The authors addressed all my concerns and I have no further comments. I support the publication of this manuscript.

Reviewer #3:

Remarks to the Author:

The authors addressed the questions and concerns raised by this reviewer. I still have two remarks regarding their answers:

1. Now I understand that the two devices presented in the main text and SI, respectively display qualitatively different behavior due to their very different relative rotation angles to the h-BN substrate (large vs. small). This, on one hand clarifies my original concern, on the other hand it means that for the desired large rotation angles, the authors have investigated only a single device. While not without precedent, it would have been more reassuring to reproduce the result with least on another similar device.

2. Regarding the contrast enhancement, I just want to emphasize it that when only a part of the image is modified (e.g., contrast enhanced) it should always be clearly marked and discussed in the figure and caption, as it is now in the revised fig. 5b.

Point by point response to the reviewers' comments:

REVIEWERS' COMMENTS

Reviewer #1 (Remarks to the Author):

During the revision, authors have significantly improved the quality of the manuscript. I need to agree with the authors that my first impression of this work was wrong. Now the paper can be published in Nature Communication.

We are happy to hear that we could clarify the issues with the first version of the manuscript and thank the reviewer for their helpful remarks and suggestion to publish the paper.

Reviewer #2 (Remarks to the Author):

The authors addressed all my concerns and I have no further comments. I support the publication of this manuscript.

We are happy to hear that we could clarify the issues with the first version of the manuscript and thank the reviewer for their helpful remarks and suggestion to publish the paper.

Reviewer #3 (Remarks to the Author):

The authors addressed the questions and concerns raised by this reviewer. I still have two remarks regarding their answers:

1. Now I understand that the two devices presented in the main text and SI, respectively display qualitatively different behavior due to their very different relative rotation angles to the h-BN substrate (large vs. small). This, on one hand clarifies my original concern, on the other hand it means that for the desired large rotation angles, the authors have investigated only a single device. While not without precedent, it would have been more reassuring to reproduce the result with at least one other similar device.

We are glad that we could sort out the issue from the previous version of the manuscript. Regarding data from more samples, we agree with the reviewer that this would be desirable. In this case, due to various constraints (first author now employed outside academia, intricate sample preparation), we could not generate more samples on short notice. On the other hand, the data are so clear (see, e.g., the perfect match of the flat band condition in Fig. 4c at one flux quantum) that they could not be explained by the fortuities of a particular sample.

2. Regarding the contrast enhancement, I just want to emphasize it that when only a part of the image is modified (e.g., contrast enhanced) it should always be clearly marked and discussed in the figure and caption, as it is now in the revised fig. 5b.

We agree that the presentation in the previous version of Figure 5b did not make contrast enhancement entirely clear. The reviewer's comment has thus helped to make the manuscript more accessible.